# Are GANs overkill for NLP?

**David Alvarez-Melis**[*]
Microsoft Research
daalvare@microsoft.com

**Vikas Garg**[*]
YaiYai Ltd and Aalto University
vgarg@csail.mit.edu

**Adam Tauman Kalai**[*]
Microsoft Research
adam@kal.ai

## Abstract

This work offers a novel theoretical perspective on why, despite numerous attempts, adversarial approaches to generative modeling (e.g., GANs) have not been as successful for certain generation tasks, particularly sequential tasks such as Natural Language Generation, as they have in others, such as Computer Vision. In particular, on sequential data such as text, maximum-likelihood approaches are significantly more utilized than GANs. We show that, while it may seem that maximizing likelihood is inherently different than minimizing distinguishability, this distinction is largely an artifact of the limited representational capacity of the model family, for a wide class of adversarial objectives. We give a theoretical model in which minimizing KL-divergence (i.e., maximizing likelihood) is a more efficient approach to effectively minimizing the same distinguishability criteria that adversarial models seek to optimize. Reductions show that minimizing distinguishability can be seen as simply boosting likelihood for certain families of models including n-gram models and neural networks with a softmax output layer. To achieve a full polynomial-time reduction, a novel next-token distinguishability model is considered. Some preliminary empirical evidence is also provided to substantiate our theoretical analyses.

## 1  Introduction

Consider a situation where one has samples from a true distribution $p$ over a set $X$ and one wishes to learn to generate similar samples, such as learning to generate English sentences from a large English text corpus. One seeks an approximation $q$ of $p$ which is "close" in some sense and from which samples can efficiently be generated. A common approach to fit these models is Maximum Likelihood Estimation (MLE), which given a training set from $p$ and a parametrized distribution $q_\theta$ seeks parameters $\theta$ that maximize the likelihood $q_\theta$ assigns to a training set. MLE has long been one of the most popular methods for fitting generative models of sequential data, such as language, where autoregressive neural language models generate remarkably realistic text, e.g., GPT-3 [6] and PaLM [11]. MLE generally involves computing likelihoods $q_\theta(x)$ which can be more challenging in some domains than others, e.g., it may be more difficult to estimate the probability of a (high-dimensional, real-valued) image than a (discrete-valued) sentence.

An alternative approach, Generative Adversarial Networks (GANs), has become popular across several domains, particularly Computer Vision, owing to breakthrough realism in the images they output [e.g., 19, 65]. GANs employ an adversarial approach to generation through a zero-sum game between a generator and a distinguisher in which the generator produces samples $x \in X$ which the distinguisher tries to distinguish from real samples from $p$. Often, both the generator and the distinguisher are differentiable neural networks, though this min-max approach of choosing a model whose outputs are nearly indistinguishable from real examples might be considered for any families of generative models and distinguishers. A major advantage of GANs (particularly for images) is that they can be used for generation without requiring computing likelihoods. This advantage is not

---

[*]equal contribution, alphabetic ordering

36th Conference on Neural Information Processing Systems (NeurIPS 2022).

significant for many sequential models such as language models, where computing likelihoods is not difficult.

In contrast, the adversarial approach has yet to demonstrate significant improvements in some other domains such as Natural Language Processing (NLP). One well-known barrier to NLP GANs is that language models produce discrete outputs (words), so they are not naturally differentiable [18]. However, despite numerous works circumventing this limitation and adapting GANs to text generation [63, 42, 23, 12], adversarial-based models have yet to achieve the same popularity or performance gains that were seen for images. In particular, language GANs have been shown to under-perform MLE in terms of quality [55] while facing the challenge of lack of diversity due to mode collapse [7], which is a well-known issue with GANs in other domains.

## 1.1 Likelihood and Distinguishability: Two sides of the same coin?

In this work, we suggest a different, fundamental barrier to adopting GANs in domains where MLE is prevalent: the adversarial approach of minimizing distinguishability can be seen as an indirect method of maximizing likelihood on observed data, and hence employing MLE directly can be more efficient. This is the case in NLP where, unlike computer vision, a measure of likelihood called *perplexity* has been the prevailing metric for training and evaluating language models for decades. We show how GANs boost likelihood in the spirit of, and inspired by, the related celebrated work of Friedman et al. [16] that showed how boosting can be viewed as an iterative approach for logistic regression.

Consider a large finite set or countably infinite set $X$ and a family $Q$ of probability distributions over $X$. For language, these might be n-gram models or neural models. Also consider a family $F$ of distinguishers $f : X \to [0, 1]$ that aim to distinguish random examples drawn from a distribution $p$ from those sampled from $q$. For any such classifier $f$, we call the difference $\alpha(f) = \mathrm{E}_q[f(x)] - \mathrm{E}_p[f(x)]$ the distinguishability *advantage* of $f$ because it quantifies the accuracy of $f$ at the task of identifying "fake" examples. A perfect distinguisher would thus have $\alpha(f) = 1$, while $f(x) = 1/2$ which predicts at random has $\alpha(f) = 0$. More formally, imagine picking $y \in \{0, 1\}$ uniformly at random and picking a random example $x$ from $q$ if $y = 1$ and from $p$ if $y = 0$. The (randomized) binary classifier that predicts, for any $x$, $\hat{y} = 1$ with probability $f(x)$, has (expected) accuracy:

$$\frac{1}{2} \sum_x q(x)f(x) + \frac{1}{2} \sum_x p(x)(1 - f(x)) = \frac{1}{2} + \frac{1}{2}\alpha(f).$$

Given a family $F$, we define the *distinguishability* of $q$ from $p$ to be $d(q) = \max_{f \in F} \alpha(f)$. Distinguishability is known to be a lower-bound on *total variation distance* (also called statistical distance), a measure of distance between distributions that is difficult to directly estimate for large domains $X$ [54]. The "Bayes-optimal" distinguisher simply predicts 1 iff $q(x) > p(x)$, and has advantage equal to the total variation distance [see, e.g., 24]. Clearly $d(p) = 0$, i.e., $p$ is indistinguishable from itself. Motivated by this observation, numerous *adversarial* approaches to approximating $q$ have been attempted to minimize distinguishability $d(q)$. If $p \in Q$ then $d(q)$ is minimized at $q = p$. We first discuss some important aspects of distinguisability.

**What adversarial objectives can be analyzed via distinguishability?** It is important to emphasize that distinguishability is indeed the objective that several adversarial approaches, such as many variants of GANs, seek to optimize. Depending on the context, it serves the purpose of *discriminator* or *critic*. In particular, as established in [54], GANs that are trained based on Kantorovich metric, Fortet-Mourier metric, dual-bounded Lipschitz distance (or the Dudley metric), total variation distance, and kernel distance can all be cast in terms of distinguishability. Thus, in particular, our results hold for GNN formulations such as Wasserstein GANs [1], MMD GANs [33], Fisher GANs [36], and Sobolev GANs [49], for an appropriately chosen family $F$ of distinguishers. For example, we obtain Wasserstein GANs, in its dual form, as a special case when $F$ is restricted to 1-Lipschitz functions in which case it can also be viewed as a special case of the so-called f-GANs [3, 37, 50]. Likewise, we obtain MMD-GANs when $F$ pertains to functions (kernels) defined over a ball in some Reproducing Kernel Hilbert Space [4]. Note that distinguishability allows us to accommodate other sophisticared GAN variants such as WGAN-GP [22, 60] that do not suffer from the issue of gradients vanishing on discrete spaces. Recall the WGAN-GP objective can be expressed in our notation as:

$$\mathrm{E}_q[f(x)] - \mathrm{E}_p[f(x)] + \lambda \mathrm{E}_r[(\|\nabla_x f(x)\|_2 - 1)^2].$$

This can be viewed as Lagrangian relaxation of the following hard objective for $\epsilon > 0$ as:

$$\mathrm{E}_q[f(x)] - \mathrm{E}_p[f(x)] \qquad \text{s.t.} \mathrm{E}_r[(||\nabla_x f(x)||_2 - 1)^2] \;\; \leq \;\; \epsilon \,.$$

Distinguishability advantage can then be readily be expressed as

$$\max_{f : \mathrm{E}_r[(||\nabla_x f(x)||_2 - 1)^2] \leq \epsilon} \mathrm{E}_q[f(x)] - \mathrm{E}_p[f(x)] \,.$$

**Is distinguishability symmetric or asymmetric?** Note that, by definition, distinguishability is asymmetric in the sense that in general $q$ *is distinguishable from* $p$ is different from $p$ *is distinguishable from* $q$. Note, however, that we recover integral probability metric (IPM) [54] when $-f \in F$ for all $f \in F$. Clearly, in this case the notion of distinguishability becomes symmetric as the advantage reduces to $\max_{f \in F} |\mathrm{E}_q[f(x)] - \mathrm{E}_p[f(x)]|$. Thus, distinguishability lets us handle an extremely wide class of discrepancies, symmetric as well as asymmetric.

**Example where maximizing likelihood $\neq$ minimizing distinguishability.** When $p \notin Q$, minimizing distinguishability among $q \in Q$ may be different than maximizing the likelihood of $q$. For instance, consider modeling the age in years of humans (say the entire population on earth) as a uniform distribution $q_m$ over $x \in \{0, 1, 2, \ldots, m\}$. Now, the $m$ which maximizes likelihood would be the age of the oldest person, which is $m = 119$ at the time of this article—any smaller $m$ would assign zero probability to the 119-year-old and thus to the entire population. However, this distribution is very distinguishable from the true distribution—for instance it assigns probability $\sim 17\%$ to being over 100 years, which is extremely unlikely among today's population. A smaller $m < 100$ would yield less distinguishable samples. While it may seem therefore that distinguishability and likelihood are inherently different criteria, as we shall see this is an artificial limitation due to the weakness of family $Q$.

Of course, the (in)equivalence depends on the families $F$ of distinguishers and $Q$ of probability distributions. We give two results showing that maximizing likelihood and minimizing distinguishability are equivalent as long as $F$ and $Q$ are similar in representational capacity, even when $p \notin Q$. First, we consider families $Q$ that are "log-linear" over some set $F$ of functions, which include n-gram models and neural networks whose top layer is a softmax, among others. The equivalence in this case is particularly simple and serves to illustrate how MLE can be a simpler way to reach the same optimum. In this case, $Q$ and $F$ are naturally paired.

**Maximizing likelihood = minimizing distinguishability for log-linear $Q$.** In the above age example, the family $Q$ of geometric distributions $q_\theta(n) \propto \exp(-\theta n)$ for $\theta > 0$ is an example of a log-linear family. We show that if $q$ can be distinguished from the population distribution $p$ by a function $f \in F$, then folding $f$ into $q$ yields a new model in $Q$ with greater likelihood. In practice, one only has a sample of the true distribution $p$ (not the entire population) and maximizing log-likelihood is approximation of minimizing KL divergence $D_{\mathrm{KL}}(p \| q) = \sum_x p(x) \log \frac{p(x)}{q(x)}$. We give formal statements about minimizing KL-divergence as well.

The conclusion of this first observation is that if a GAN were to converge within a log-linear family (and making GANs converge is often not an easy feat in practice), it would converge to the MLE.

**General polynomial-time reduction.** Our second result is a polynomial-time reduction from likelihood maximization to next-token distinguishability, without the log-linear requirement. We consider the common case of (unidirectional) sequential models that predict the next token based on the previous tokens, which have several practical advantages including being efficient to compute—the probability of a sequence is simply the product of the conditional probabilities of each subsequent word given the previous words. Many state-of-the-art transformer language models such as GPT-3 take this form. Achieving an efficient reduction is challenging due to the normalization requirement of computing partition functions. In order to achieve a polynomial-time reduction, we consider a notion of next-token distinguishability, where the game is as follows: a prefix of tokens is chosen, based on which the generator generates a token to follow the prefix. Given the actual next token and the generated next token, the distinguisher aims is to identify which is which. Algorithm 1 leverages a next-token distinguisher to iteratively increase likelihood. In particular, given any target $\epsilon > 0$, Theorem 1 shows that Algorithm 1 will terminate and output an efficiently computable model $q$ which is nearly (to within $\epsilon$) indistinguishable from the truth, and it runs in time polynomial in $1/\epsilon$.

If $p \in Q$ and one has an optimal distinguisher, one will eventually converge to a model close to $p$, as has been discussed heavily in the literature. However, our results are also meaningful in the more realistic case where one has imperfect distinguishers.

**Contributions**. The main contributions of this paper are:

- showing that, although in general minimizing distinguishability and maximizing likelihood seem to be different, they are in fact closely related,
- introducing a new model of next-token distinguishability that is necessary to make the reduction efficient, and
- offering a new perspective on why GANs might have been less successful in NLP and other sequential domains as they have been, e.g., for images.

**Organization**. We begin by summarizing related work on GANs, especially for text. We then illustrate how GANs can be overkill for the simple case of n-gram models in Section 3. Section 4 covers log-linear models. Section 5 gives explicit bounds on general reductions between maximizing likelihood and minimizing distinguishability. Section 6 shows how the reduction can be efficiently computed in the case of sequential inputs, from which we propose a simple polynomial time algorithm that provably finds a distribution which is nearly-indistinguishable with respect to a given class of discriminator functions. Finally, we discuss the relevance of our work in Section 8. All proofs are deferred to the Appendix.

## 2   Related Work

**Generative models**. Several approaches to generative modeling have been investigated, especially in the context of images. In particular, impressive results have been obtained recently with variational autoencoders, GANs, normalizing flows, autoregressive models, diffusion processes, and score/energy based models [28, 19, 41, 40, 25, 53, 59, 10]. Generally, training approaches are either adversarial; or rely on MLE, contrastive divergence estimation, or score matching [52]. Some connections have begun to emerge between these models, and alternate training procedures have been advocated [9, 53, 62].

**A word of caution:** Understanding the theoretical underpinnings of generative models with respect to their sample quality is an intriguing question that has been previously investigated by several seminal works such as [27, 21, 39, 56] and requires further analyses. Our objective here is not to claim at all that maximizing likelihood is universally better than adversarial methods or vice-versa, but to emphasize that for many problems, in domains like NLP, the two objectives often turn out to be equivalent *mathematically* via the notion of distinguishability and maximizing MLE could provide a more efficient (and stable way) of optimizing the common objective. Also, note that maximizing likelihood does not always correlate with the perceptual sample quality [21]. A more comprehensive analysis encompassing the effect of architecture, optimization procedures and issues such as trade-off between perception and distortion [5], training and inference time, mode-collapse etc. is required for better understanding of generative models. In that sense, our message is consistent with observations made previously [39, 56, 21] that there is no single model that fits all situations, and the *right* model depends on the specific requirements of applications.

**GANs for text**. Since their introduction [19], there has been interest in adapting GANs to text data. The driving motivation was that —up until very recently— samples generated by traditional (likelihood-based) models had been easy to distinguish from human-generated text, and the success of image GANs at generating realistic-looking samples suggested a possible avenue to improve the quality of their natural language counterparts.

The first and most significant challenge in adapting GANs to text arises from the very nature of this data. Goodfellow [18] points out that GANs require the generator to be differentiable, which poses a challenge for discrete text representations such as one-hot word or character representations. Two of the most popular approaches to circumvent this obstacle are policy gradient techniques (e.g., REINFORCE [61]) —which when applied to language modeling nevertheless often require maximum likelihood pre-training [8, 63])— and the Gumbel-Softmax approximation [30]. The few adversarial methods that do not require pre-training (e.g., [42, 45]) have failed to show significant promise in all but a few artificial tasks.

This nascent but active line of work seemed to suggest for a period of time that GANs might provide a breakthrough in text generation. This promise did not fully materialize, and instead the most recent breakthrough came from models building very large transformer-based architectures like GPT [43, 44, 6] or PaLM [11] — which are trained with traditional cross-entropy (MLE) objectives.

Yet the question of how GAN-based methods for text compare with likelihood-based ones still garners significant interest, and while various works have provided an empirical comparison between them —with most of these suggesting the advantage of MLE-based ones [7]— theoretical explanations have been less explored.

**Relating objectives via divergences**. The connection between maximum likelihood estimation, distinguishability and divergences between probability distributions has been explored before. For example, it is well known that maximizing likelihood is equivalent to minimizing the KL divergence between certain families of fitted and reference distributions, though this is not the only divergence for which such a connection exists [46]. On the other hand, from the moment GANs were introduced, Goodfellow et al. [19] noted that —assuming a perfect discriminator— the adversarial objective corresponds to minimizing a Jensen-Shannon divergence. Furthermore, the minimal discrimination error is also directly related to the total variation distance (see, e.g., Hashimoto et al. [24]). On the other hand, for exponential families the gradient of the KL divergence is known to be related to the discrepancy between distributions [57]. While conceptually similar to this line of work, here instead we give an *explicit* reduction that shows how distinguishability and (log) likelihood are in direct correspondence.

Pinsker's inequality is a well-known result linking KL divergence and total variation distance (TVD): $\text{TVD} \leq \sqrt{\text{KL}/2}$. While related, this inequality is not directly relevant to the context of this work. First, while total variation provides an upper bound to distinguishability, it is not computable in general, so it is rarely used as a training objective for generative models. On the other hand, being one-sided,[2] it does not imply that reducing TVD reduces KL divergence. Furthermore, Pinsker's is in general a very loose inequality, particularly for the direction of KLD that is equivalent to MLE (i.e., $D_{\text{KL}}(p \parallel q_\theta)$), since if $p(x) > 0 \approx q_\theta(x)$ even for a single $x$ leads to unbounded KL divergence. In contrast, in this work we provide a *direct reduction* directly linking the two criteria of interest: distinguishability and maximum likelihood.

**Log-linear language models**. In this work we focus our analysis on log-linear models [31, 34], which are widely used in natural language processing (often known in that community as Maximum Entropy –MaxEnt– models) for various tasks. In particular, these models have been a cornerstone of both neural [2, 35] and non-neural [47, 26] language modeling.

**Boosting**. The reduction shown here bears resemblance to boosting. It is well-known that boosting can be analyzed through the lens of maximum likelihood (Friedman et al. [16]), while Lebanon and Lafferty [32] formalized the equivalence of AdaBoost and maximum likelihood training for exponential models. More recently, boosting has been applied to generative adversarial models [58, 20], albeit with a different approach and objective than the connection drawn in this work.

## 3   Illustration: GANs for n-gram language models

To illustrate our main point, consider first the simplest model of language: a unigram model where the probability of each word is independent, and the end of sentence token EOS has a given probability as well. If $\theta_w$ represents the log-probability of word $w$, then the log-probability of sentence $w_1 \ldots w_t$ is given by:

$$\log q(w_1 w_2 \ldots w_t) = \theta_{w_1} + \theta_{w_2} + \ldots + \theta_{w_t} + \theta_{\text{EOS}}.$$

The MLE parameters $\theta^*$ can be computed in linear time by simply counting word frequencies.

A more roundabout approach to fitting a unigram language model would be to start with any initial unigram model $q$, generate random samples from $q$ and compare them to those from $p$. One could then distinguish the two by finding a word that appears significantly more often in one than in the other. For example, if one generates text from the model $q$ and finds that the word "the" occurs much more often in text generated from $p$, one would then update the parameters by increasing $\theta_{\text{the}}$ (and

---

[2]Reverse Pinsker's inequalities exist only for particular cases, but they too are very loose in general [48].

decreasing $\theta_{w'}$ for all other words $w'$ to keep $q$ a probability distribution). As we shall see later, if this more involved procedure converged, it would necessarily converge to the same maximum-likelihood estimator $\theta^*$.

A similar argument applies to any $n$-gram model in which the probability of each subsequent word is determined only by the previous $n-1$ words. This is also optimized by frequency counts (a variety of "smoothing" techniques, e.g., adding 1 to counts, also known as Laplace Smoothing [17, 29] are often used as a form of regularization on top of these counts). Distinguishers could similarly be used to find a model $q$ that is indistinguishable from $p$ according to $n$-gram frequencies, but again this would simply converge to the MLE parameters.

## 4 Equivalence for log-linear models

In this section, we show that there is one optimal log-linear model that both minimizes distinguishability and maximizes log-likelihood. Consider a log-linear model with features $f : X \to [0,1]^d$, i.e., $d$ bounded features $f_i : X \to [0,1]$. The model predicts

$$q_\theta(x) = \frac{\exp\left(\langle f(x), \theta \rangle\right)}{Z_\theta} \ , \tag{1}$$

where $\langle \cdot, \cdot \rangle$ denotes inner product, $\theta \in \mathbb{R}^d$ is a parameter vector and $Z_\theta = \sum_x \exp\langle \theta, f(x) \rangle$ is a normalizing constant called the partition function.

In the unigram example, the features $f_i$ would be word counts normalized by dividing by the maximum allowed sentence length (to ensure $f_i(x) \le 1$). In a neural network the features $f_i$ would correspond to the top layer and $q$ computes a softmax. Multiple strategies have been studied for computing or estimating the partition function $Z_\theta$ [see, e.g., 14].

As discussed earlier, these feature functions can also be thought of as classifiers that distinguish examples drawn from $p$ from those drawn from $q_\theta$ and the advantage of $f_i$ is $\alpha(f_i) = \sum_x f_i(x)(q_\theta(x) - p(x))$. The advantage vector is $\alpha(f) = \langle \alpha(f_i) \rangle_{i=1}^d$. Note that a negative advantage can be used for distinguishing by using the reverse classifier $1 - f_i$ as a distinguisher, which has opposite advantage $\alpha(1 - f_i) = -\alpha(f_i)$.

**Observation 1.** *The gradient of $D_{KL}(p \parallel q_\theta)$ with respect to $\theta$ is the advantage vector $\alpha(f)$, i.e., for all $i = 1, 2, \ldots, d$:*

$$\frac{\partial D_{KL}(p \parallel q_\theta)}{\partial \theta_i} = \sum_x f_i(x)(q_\theta(x) - p(x)) = \alpha(f_i).$$

The above straightforward calculation is well-known as is the fact that $D_{KL}(p \parallel q_\theta)$ is convex in $\theta$. However, we interpret this fact in the context of GANs: searching for $\theta$ which gives a zero-gradient for KL divergence is equivalent to finding $\theta$ which is indistinguishable with respect to each $f_i$. While a number of GANs have be designed in various architectures that solve the seemingly more complex problem of $\min_\theta d(q_\theta)$, it can generally be more efficient to maximize likelihood, which (approximately) minimizes the KL divergence.

## 5 Distinguishability is equivalent to increasing likelihood for general $F$, $Q$

In this section, we show how reducing log-loss is equivalent to distinguishing real and generated examples. This is the basis behind a single step of our main algorithm (the reduction in this section is efficient for a single step, but the increase in runtime would lead to a general exponential-time algorithm). The bounds here are in terms of log-loss, as measured on a specific sample, rather than the abstract KL divergence quantity of the previous section, which cannot be computed exactly using a finite sample. In particular, we show how, if one can distinguish a given distribution from the sample, then one can decrease that distribution's log-loss, and vice versa.

For the remainder of this section, we drop $\theta$ from the variable denoting the fitted distribution $q_\theta$ to avoid cluttering the notation. Fix a sample $\mathcal{S} = \langle x_1, \ldots, x_n \rangle \in X^n$ of $n$ training examples drawn from $p$, and define the log-loss to be:

$$\hat{L}(q; \mathcal{S}) = -\frac{1}{n} \sum_{i=1}^n \log q(x_i) = -\hat{\mathrm{E}}_{\mathcal{S}}[\log q(x)],$$

where we use hat on $L$ to denote that the loss is estimated on a (finite) training set $\mathcal{S}$. Likewise, $\hat{\mathrm{E}}_{\mathcal{S}}[g(x)]$ denotes the empirical expectation $\frac{1}{n} \sum_{i=1}^{n} g(x_i)$. Note that the expected log-loss over training sets is known as the cross-entropy

$$H(p, q) = \mathrm{E}_{\mathcal{S} \sim p^n}[\hat{L}(q; \mathcal{S})],$$

and hence the expected difference in log-loss between two candidate distributions $q$ and $q'$ is equal to the difference

$$\mathrm{E}_{\mathcal{S} \sim p^n}[\hat{L}(q; \mathcal{S}) - \hat{L}(q'; \mathcal{S})] = D_{\mathrm{KL}}(p \parallel q) - D_{\mathrm{KL}}(p \parallel q'),$$

so minimizing log-loss approximately minimizes the KL divergence. Also, we define the *training advantage* of distinguisher $f : X \to [0, 1]$ to be:

$$\hat{\alpha}(f) = \mathrm{E}_q[f(x)] - \hat{\mathrm{E}}_{\mathcal{S}}[f(x)] \tag{2}$$

which is independent of $p$, depending on the sample alone and can thus be estimated to arbitrary accuracy using samples generated from $q$. The lemmas below show how one can use a distinguisher to reduce log-loss on the same training sample, and how to use a distribution with a lower log-loss to distinguish the two distributions.

**Lemma 1.** *Let $a \geq 0$ and suppose $f : X \to [0, 1]$ has training advantage $\hat{\alpha}(f) \geq a$. Then, the probability distribution $q'(x) = q(x)e^{-af(x)}/Z_{q'}$ where $Z_{q'} = \sum_x q(x)e^{-af(x)}$, has lower log-loss:*

$$\hat{L}(q'; \mathcal{S}) \leq \hat{L}(q; \mathcal{S}) - a^2/2.$$

Before we give the proof, we note that if $f$ is computed by a neural network and $q$ is computed as a neural network with a softmax at the top layer, i.e., $q(x) = e^{\langle v, g(x) \rangle} / \sum_x e^{\langle v, g(x) \rangle}$ where $g : X \to \mathbb{R}^d$ is some neural network, then $q'$ is naturally represented as the combined neural network with softmax $q'(x) \propto e^{\langle (v, -a), (g(x), f(x)) \rangle}$ in $d + 1$ dimensions.

This means that if we can distinguish $\mathcal{S}$ from $q$, then we can simply reduce log-loss by down-weighting suspicious samples that satisfy the distinguisher $f(x)$. The difference between this statement and Observation 1 is analogous to the difference between boosting and logistic regression [16]. In logistic regression, one typically fixes the set of features in advance, whereas in boosting this is not possible if there are infinitely many possible classifiers.

Conversely, we next show that if $q'$ has a lower log-loss than $q$ on the training samples, then we can distinguish $q$ from these samples.

**Lemma 2.** *For any constant $C > 1$ and distributions $q, q'$ such that $\frac{1}{C}q(x) \leq q'(x) \leq Cq(x)$ for all $x \in X$, the distinguisher $f : X \to [0, 1]$ defined by,*

$$f(x) = \frac{1}{2 \log C} \log \frac{Cq(x)}{q'(x)},$$

*has a training advantage of,*

$$\hat{\alpha}(f) \geq \frac{\hat{L}(q) - \hat{L}(q')}{2 \log C}.$$

Importantly, due to the logarithmic dependence on $C$, the above lemma is meaningful even if $q$ and $q'$ are exponentially far apart so long as they have the same support.

Lemma 1 implies a reduction between the problem of distinguishing with nontrivial advantage to non-trivially reducing log-loss for log-linear families. Note that iteratively applying the reduction requires repeated computation of the normalization terms over $X$, and computing such partition functions is an area of active research—where it is known how to do it efficiently for some classes and not for others. The next section gives an efficient reduction for (unidirectional) sequential models.

## 6    Efficient Reduction for Sequential Models

This section gives an efficient reduction from distinguishing to MLE for sequential models. This requires showing how one can efficiently compute the normalization terms (partition function) on a token-by-token basis for black-box sequential (e.g., language or auto-regressive) models. The

key insight for efficiency is that, rather than distinguishing entire sequences from $p$ and $q$, one distinguishes the conditional next-token predictions. In particular, rather than generating entire sequences from $q$, one can generate next-token predictions on all sequence prefixes in the training data.

Clearly, evaluating a neural network over all sequences is infeasible. However, in many applications such as NLP, the inputs are sequential $x = (x_1, \ldots, x_\ell)$, where every token $x_i$ is taken from a large discrete vocabulary. In such cases, the combinatorial nature of the data makes density estimation intractable unless the likelihood computation is broken into small sequential steps by representing the overall probability as the $\Pr(x) = \prod_j \Pr(x_j | x_1, x_2, \ldots x_{j-1})$.

In this section we show how a natural extension of the framework described above allows us to achieve an efficient reduction for this common type of sequential model. To do so, we define a simple generalization of the training advantage criterion (2), which now relies on a *step-wise distinguisher $g$* operating on variable-length sequences. Formally, we consider a *language* of $N$-length sequences[3] of tokens taken from a vocabulary $\mathcal{V}$, and distinguisher functions $g : \bigcup_{j=1}^{N} \mathcal{V}^j \to [0, 1]$, i.e., functions which can take subsequences of any size as input. Given a sample $\mathcal{S}$ of sequences, we say that $g$ has *generalized training advantage* given by

$$\hat{\beta}(g) = \hat{\beta}(g, \mathcal{S}, q) = \frac{1}{N} \sum_{j=1}^{N} \hat{\mathbb{E}}_{x \sim \mathcal{S}} \left[ \mathbb{E}_{w \sim q(\cdot | x_0, \ldots, x_{j-1})} g(x_0, \ldots, x_{j-1}, w) - g(x_0, \ldots, x_j) \right] \quad (3)$$

where, by convention, $x_0 = \emptyset$, so that $q(x_0, w) = q(w)$. This criterion can be interpreted as follows. For every length $j \in \{1, 2, \ldots, N\}$, $g$ is tasked with distinguishing a subsequence consisting of the first $j$ tokens in a *true* sequence sampled from $\mathcal{S}$ from another $j$-length sequence in which the last element is replaced by a randomly selected token from the alternate distribution $q$.

**Lemma 3.** *Let $b \geq 0$ and suppose $g : \bigcup_{j=1}^{N} \mathcal{V}^j \to [0, 1]$ has generalized training advantage $\hat{\beta}(g) \geq b$. We define a distribution $q'$ through its conditional probabilities as:*

$$q'(x_j \mid x_1, \ldots, x_{j-1}) = q(x_j \mid x_1, \ldots, x_{j-1}) e^{-bg(x_1, \ldots, x_j)} / Z_{q'}(x_1, \ldots, x_{j-1})$$

*where now $Z_{q'}(x_1, \ldots, x_{j-1}) = \sum_{\tilde{x}_j} q(\tilde{x}_j \mid x_1, \ldots, x_{j-1}) e^{-bg(x_1, \ldots, x_{j-1}, \tilde{x}_j)}$. Then $q'$ incurs lower log-loss than $q$:*

$$\hat{L}(q'; \mathcal{S}) \leq \hat{L}(q; \mathcal{S}) - Nb^2/2.$$

The proof is deferred to Appendix C.

Next, we use Lemma 3 repeatedly to derive a simple algorithm that, given access to non-trivial weak distinguishers, returns a distribution that is nearly indistinguishable (by that class) from the true distribution $p$. Formally, let $\mathcal{G} = \{g \mid g : \bigcup_{j=1}^{N} \mathcal{V}^j \to [0, 1]\}$ be a class of distinguishers. We assume access to an oracle $O_d : \mathcal{Q} \mapsto \mathcal{G}$ which for any $q \in \mathcal{Q}$ returns a distinguisher $g$. In practice, such as in typical GAN training setting, one could think of this oracle as being approximated by the subroutine that trains the discriminator. We say that $q$ is $\epsilon$-indistinguishable by oracle $O_d$ if its output $g$ has advantage $\hat{\beta}(g, \mathcal{S}, q) \leq \epsilon$. We do not need to assume that $O_d$ is optimal in any sense.

---

**Algorithm 1:** Boosted weak distinguishers.

---

**Input:** Initial model $q_0$, corpus $\mathcal{S}$, distinguisher oracle $O_d$, advantage threshold $\epsilon$.
$t \leftarrow 0$
**while** *True* **do**
    $g_t \leftarrow O_d(q_t)$
    $b_t \leftarrow \hat{\beta}(g_t, \mathcal{S}, q_t)$
    if $b_t < \epsilon$, **Output:** $q_t$
    Compute $q_{t+1}(x) \triangleq q_t(x) e^{-b_t g_t(x)} / \sum_{x \in \mathcal{S}} q_t(x) e^{-b_t g_t(x)}$ on entire corpus $\mathcal{S}$
    $t \leftarrow t + 1$
**end**

---

[3]Padding can be used to handle sequences of variable length.

**Theorem 1.** *Let $q_0$ be a language model and let $\epsilon > 0$. Algorithm 1 returns a distribution $q^*$ which is $\epsilon$-indistinguishable from $\mathcal{S}$ by oracle $\mathrm{O}_d$. It runs in $O\big(\frac{1}{\epsilon^2}L_0(\frac{T_d}{N} + nT_g(m+n))\big)$ time, where $L_0 = \hat{L}(q_0; \mathcal{S})$ is the log-loss of $q_0$, $T_d$ is the runtime of oracle $\mathrm{O}_d$, $T_g$ is the complexity of evaluating any distinguisher $g$ on a single input, $n = |\mathcal{V}|$ is the vocabulary size, $N$ is the sequence length and $m = |\mathcal{S}|$ is the number of training sequences.*

Thus, Algorithm 1 combined with Theorem 1 and Lemma 3 yields a polynomial-time reduction from distinguishing distributions to maximum likelihood estimation for sequential models.

## 7 An empirical validation of the reductions

We provide empirical validation of Lemma 3. We train a pair of text generator and discriminator using a publicly available implementation[4] of SeqGAN [63]. The generator is pretrained by negative log-likelihood (NLL) minimization. During the adversarial phase of training, the generator is trained using policy gradient. After training, we compute the discriminators' generalized training advantage (Equation 2), using finite-sample empirical approximations), and then create a new generator whose next-word logit predictions are modified according to Lemma 3. We compare the NLL of the original and 'boosted' generators across training epochs, and compare the difference between these to the theoretical lower bound of Lemma 3.

The results, shown in Figure 1 in the Supplement, correspond to the default SeqGAN settings in terms of network capacities and language configuration (maximum sequence length=20, vocabulary size=5000). These results show that the 'boosting' underpinning Lemma 3 does indeed improve likelihood (reduces NLL) as stated, and that the empirical difference is indeed lower-bounded by the gap predicted by theory.

## 8 Discussion and conclusions

In this work, we have argued that minimizing log-loss (i.e., KL-divergence) and minimizing statistical distinguishability are tightly related goals. Specifically, if the families of distinguishers and probability distributions are of similar power, then one can use a distinguisher to reduce log-loss. This is the case, e.g., for n-gram language models (and other sequential tasks), for which perplexity (a measure of likelihood) is easy to compute, naturally meaningful, and allows for efficient sampling. Indeed, for a long time, minimizing log-loss has been the objective with which most state-of-the-art models have been trained. For such models, Lemma 1 implies that if one can distinguish the model from samples by a neural network then one can construct a larger neural network with lower log-loss. Hence, one may prefer to simply train a larger model in the first place for some applications.

**Text vs. images.** Note that we often can compute conditional likelihoods more efficiently for text data compared to images. The sequential nature of text data allows us to compute the normalization terms (and thus the partition function) on a token-by-token basis, thereby enabling us to distinguish the conditional next-token predictions instead of having to distinguish full sentences. Similarly, for low dimensional images (such as 8x8, or 4x4), conditional predictions are tractable and thus autoregressive modeling allows for efficient training and sampling. In contrast, MLE based autoregressive models are typically slow for high dimensional real images. In such settings estimating the partition function is challenging, so alternative methods such as noise contrastive estimation, score matching, Langevin dynamics, and MCMC sampling in latent space that exploit connections between GANs and energy based models have been preferred [38, 13, 64, 15, 9, 51]. We hope this work fosters further research on the comparative aspects, both pros and cons, of different generative approaches for different NLP and vision applications.

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
