# A  Proof of Lemma 1

*Proof (Lemma 1).*

$$
\begin{aligned}
\hat{L}(q;\mathcal{S}) - \hat{L}(q';\mathcal{S}) &= \hat{\mathrm{E}}_{\mathcal{S}}[\log q'(x) - \log q(x)] \\
&= \hat{\mathrm{E}}_{\mathcal{S}}\left[-af(x) - \log Z_{q'}\right] \\
&= a(\mathrm{E}_q\left[f(x)\right] - \hat{\mathrm{E}}_{\mathcal{S}}\left[f(x)\right]) - a\mathrm{E}_q\left[f(x)\right] - \log Z_{q'} \\
&= a\hat{\alpha}(f) - a\mathrm{E}_q\left[f(x)\right] - \log Z_{q'} \tag{4}
\end{aligned}
$$

Since $\hat{\alpha}(f) \geq a$ by assumption, it remains to show $a\mathrm{E}_q\left[f(x)\right] + \log Z_{q'} \leq a^2/2$. Using the bound $\log r \leq r - 1$ for any $r > 0$, we get that,

$$
\begin{aligned}
a\mathrm{E}_q\left[f(x)\right] + \log Z_{q'} &\leq a\mathrm{E}_q\left[f(x)\right] + Z_{q'} - 1 \\
&= a\mathrm{E}_q\left[f(x)\right] + \mathrm{E}_q[e^{-af(x)}] - 1 \\
&= \mathrm{E}_q\left[af(x) + e^{-af(x)} - 1\right] \\
&\leq \mathrm{E}_q[(af(x))^2/2],
\end{aligned}
$$

where we have used the fact that $Z_{q'} = \mathrm{E}_q[e^{-af(x)}]$ and, to get to the last line we use $e^{-r} + r - 1 \leq r^2/2$ for $r \geq 0$ by Taylor expansion. Since $f(x) \in [0,1]$, the last quantity is at most $a^2/2$, which together with (4), gives $\hat{L}(q;\mathcal{S}) - \hat{L}(q';\mathcal{S}) \geq a^2/2$. $\quad\square$

# B  Proof of Lemma 2

*Proof (Lemma 2).* Let $g(x) = \log q(x) - \log q'(x)$. By Jensen's inequality,

$$
\begin{aligned}
\mathrm{E}_q[g(x)] - \hat{\mathrm{E}}_{\mathcal{S}}[g(x)] &= -\mathrm{E}_q\left[\log \frac{q'(x)}{q(x)}\right] - \hat{\mathrm{E}}_{\mathcal{S}}[g(x)] \\
&\geq -\log \mathrm{E}_q\left[\frac{q'(x)}{q(x)}\right] - \hat{\mathrm{E}}_{\mathcal{S}}[g(x)] \\
&= -\log(1) - \hat{\mathrm{E}}_{\mathcal{S}}[g(x)] \\
&= \mathrm{E}_{\mathcal{S}}[\log q'(x)] - \hat{\mathrm{E}}_{\mathcal{S}}[\log q(x)] \\
&= \hat{L}(q;S) - \hat{L}(q';S)
\end{aligned}
$$

Since $f(x) = \frac{1}{2\log C}(g(x) + \log C)$, the training advantage of $f$ is that of $g$ scaled by a factor of $\frac{1}{2\log C}$. Finally, it is straightforward to verify that $f(x) \in [0,1]$ by our assumptions on the ratio between $q$ and $q'$. $\quad\square$

# C  Proof of Lemma 3

*Proof.* We proceed analogously as in the proof of Lemma 1. We first note that

$$
\hat{L}(q;\mathcal{S}) = -\hat{\mathrm{E}}_{\mathcal{S}}\left[\log \prod_{i=1}^{N} q(x_i \mid x_1, \ldots, x_{i-1})\right] = -\sum_{i=1}^{N} \hat{\mathrm{E}}_{\mathcal{S}} \log q(x_i \mid x_1, \ldots, x_{i-1}),
$$

and

$$
\begin{aligned}
\hat{L}(q';\mathcal{S}) &= -\hat{\mathrm{E}}_{\mathcal{S}}\left[\log \prod_{i=1}^{N} q'(x_i \mid x_1, \ldots, x_{i-1})\right] \\
&= \sum_{i=1}^{N} -\hat{\mathrm{E}}_{\mathcal{S}} \log q(x_i \mid x_1, \ldots, x_{i-1}) + bg(x_1, \ldots, x_i) + \log Z_{q'}(x_1, \ldots, x_{i-1})
\end{aligned}
$$

Let us use the short-hand notation $x_{1:i} \triangleq (x_1, \ldots, x_i)$. Subtracting the two equalities above we obtain

$$\hat{L}(q; \mathcal{S}) - \hat{L}(q'; \mathcal{S}) = \sum_{i=1}^{N} \hat{\mathrm{E}}_{\mathcal{S}} \left[ -bg(x_{1:i}) - \log Z_{q'}(x_{1:i-1}) \right],$$

which, after adding and subtracting $\hat{\mathrm{E}}_{\mathcal{S}} \mathrm{E}_{w \sim q(w|x_{1:i-1})} g(x_{1:i-1}, w)$ and rearranging terms, yields

$$\hat{L}(q; \mathcal{S}) - \hat{L}(q'; \mathcal{S}) = b \left[ \sum_{i=1}^{N} \hat{\mathrm{E}}_{\mathcal{S}} \left( \mathrm{E}_{w \sim q(w|x_{1:i})} g(x_{1:i-1}, w) - g(x_{1:i}) \right) \right] \tag{5}$$

$$- \sum_{i=1}^{N} \hat{\mathrm{E}}_{\mathcal{S}} \left[ \log Z_{q'}(x_{1:i-1}) - b \mathrm{E}_{w \sim q(w|x_{1:i-1})} g(x_{1:i-1}, w) \right] \tag{6}$$

$$= bN\hat{\beta}(g) - \sum_{i=1}^{N} \hat{\mathrm{E}}_{\mathcal{S}} \left[ b \mathrm{E}_w g(x_{1:i-1}, w) + \log Z_{q'}(x_{1:i-1}) \right] \tag{7}$$

By assumption we have $Nb\hat{\beta}(g) \geq Nb^2$, so it it remains to show that the second term is upper bounded by $Nb^2/2$. Using, as before, the bound $\log r \leq r - 1$ for every $r = Z_{q'}(x_{1:i-1}) \geq 0$, we get that, for every $i = 1, \ldots, N$:

$$\hat{\mathrm{E}}_{\mathcal{S}} \left[ b \mathrm{E}_w g(x_{1:i-1}, w) + \log Z_{q'}(x_{1:i-1}) \right] \leq \hat{\mathrm{E}}_{\mathcal{S}} \left[ b \mathrm{E}_w g(x_{1:i-1}, w) + Z_{q'}(x_{1:i-1}) - 1 \right]$$

$$= \hat{\mathrm{E}}_{\mathcal{S}} \left[ b \mathrm{E}_w g(x_{1:i-1}, w) + \mathrm{E}_w e^{-bg(x_{1:i-1}, w)} - 1 \right]$$

$$= \hat{\mathrm{E}}_{\mathcal{S}} \mathrm{E}_w \left[ bg(x_{1:i-1}, w) + e^{-bg(x_{1:i-1}, w)} - 1 \right]$$

$$\leq \hat{\mathrm{E}}_{\mathcal{S}} \mathrm{E}_w \left( bg(x_{1:i-1}, w)/2 \right)^2 \leq \frac{b^2}{2}$$

where the last inequality follows again from the fact that $g(x) \in [0, 1]$ for any $x$. Therefore, the sum over these $N$ terms is upper bounded by $N\frac{b^2}{2}$, which combined with (7), yields the desired result. □

## D  Proof of Theorem 1

*Proof.* The fact that Algorithm 1 terminates with a distribution $q$ which is $\epsilon$-indistinguishable by $\mathrm{O}_d$ is immediate from the stopping criterion.

Now, for the runtime analysis, note that —by construction— the iterates $g_t$, $t \in \{0, \ldots, T-1\}$ have training advantage $\hat{\beta}(g_t, \mathcal{S}, q_t) \geq \epsilon$. Thus, by Lemma 3, the algorithm makes at least $\frac{N\epsilon^2}{2}$ improvement in each iteration. Therefore, the total number of iterations $T$ is at most $\frac{2L_0}{N\epsilon^2}$, where $L_0 := \hat{L}(q_0; \mathcal{S})$ is the log-loss of the initial model. Each iteration of Algorithm 1 requires calling $\mathrm{O}_d$ oracle once, evaluating $\hat{\beta}(\cdot)$ at an $\mathcal{O}(NmnT_g)$ complexity, and updating each of the $n$ next-token probabilities of $q$ for each sequence length $1, \ldots, N$. Each of these updates involves evaluating $g$ plus an $\mathcal{O}(n)$ partition normalization. Putting these together, we conclude that each iteration has $O(T_d + NnT_g(m + n))$ complexity.

Combining the the two arguments above, we conclude that Algorithm 1 has a total runtime of $O\left(\frac{1}{\epsilon^2} L_0(\frac{T_d}{N} + nT_g(m + n))\right)$. □

## E  Empirical validation

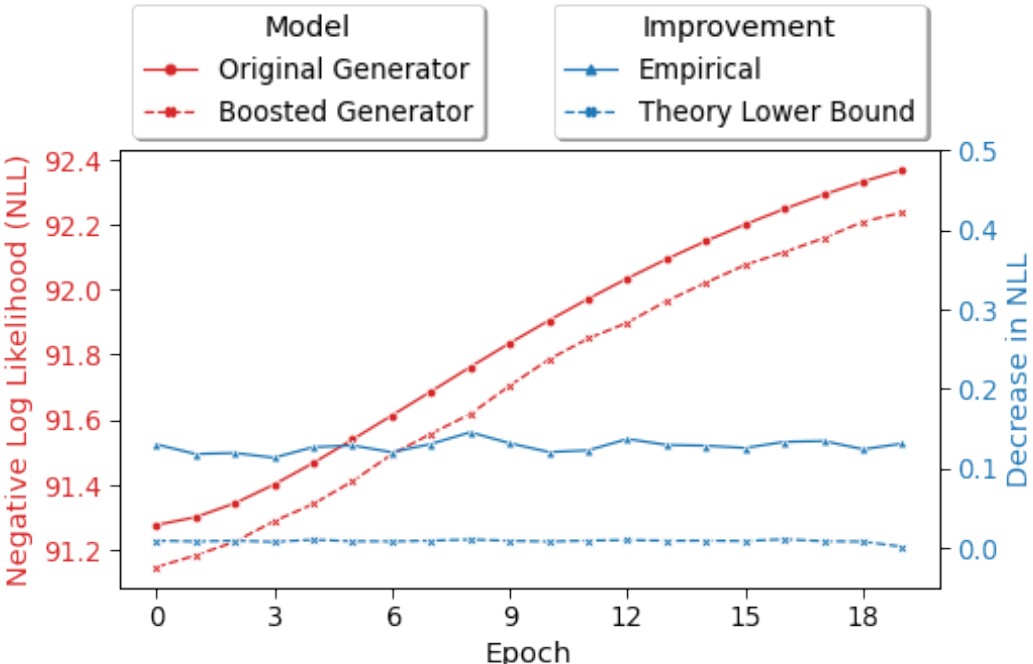

Figure 1: **Empirical Validation of Lemma 3**. For a simple pre-trained SeqGAN model (generator + discriminator), we show that the boosting scheme proposed in that Lemma results in reduced log-loss (NLL) throughout training. Furthermore, the empirical difference between original and boosted models is indeed lower-bounded by the gap predicted by the Lemma.