# OpenReview forum: "Are GANs overkill for NLP?"
_NeurIPS.cc/2022/Conference — NeurIPS 2022 Accept_

### Official Review · Reviewer_UUT1 · 2022-07-10

**Rating:** 6
**Confidence:** 3
**Soundness:** 3 good
**Presentation:** 3 good
**Contribution:** 3 good

**Summary:**

In this paper, the authors show maximizing likelihood is effectively minimizing distinguishability for log-linear Q. Authors suggest a polynomial-time reduction from likelihood maximization to next-token distinguishability. If the distinguisher has an advantage over a threshold, then a generator network with lower log-loss can be constructed accordingly. Since in sequential domains (eg, text) the likelihood is easier to compute, one might prefer to use MLE in the first place.

**Questions:**

Q1:

What’s the difference between “distinguisher” and “discriminator”? If they’re the same, maybe authors should unify the terms.

Q2:

It’s known that MLE minimizes KL divergence and GAN minimizes JS divergence, but this paper assumes GAN minimizes distinguishability. On line 70, “Motivated by this observation, numerous adversarial approaches to approximating q have been attempted to minimize distinguishability d(q)”. Is it an observation, assumption or statement? Can you prove that GAN is indeed minimizing distinguishability (e.g., with what kind of loss for generator/discriminator, using what kind objective, and give the optimal value for D)?

Q3:

When you say “distinguishable” or “distinguishability”, is it symmetrical (like JS) or asymmetrical (like KL)? Do “q is distinguishable from p” and “p is distinguishable from q” mean the same thing?

If it’s asymmetrical (like KL), then it’s not surprising that maximizing likelihood and minimizing distinguishability yield the same convergence in most cases. But I don’t think GANs are using an asymmetrical objective (because discriminator treats real and fake samples equally).

If it’s symmetrical, then the example about ages you give in line 72-82 does not seem right to me. “A smaller m < 100 would yield less distinguishable samples”. Any q with m < 100 assigns zero probability to ages over 100, hence real samples (with 100+ ages) are very distinguishable from samples in q.

Actually in the ages example, MLE(KL) and GAN(JS) both yields m=119 as the optimal point.

Q4:

Does your argument work on next-token level GAN only? By next-token level, I mean the generator takes sentence prefix as input, and output the distribution for the next token, and the discriminator takes in prefix + next_token, and predicts a score for only the next_token (assuming prefix is always real).

Another set of GAN is sentence-level, where discriminator takes in a whole sentence and predicts only one score. This kind of GAN suffers from sparse reward (with RL) or sampling difficulties.

On next-token level, it’s no news GAN is almost identical as MLE. So the conclusion is not surprising. However the mathematics and proofs in the paper is valuable.

Q5:

I still wonder what makes the fundamental difference between GAN in text and vision. Is it because text models are sequential (so likelihood is easy to compute)? Let’s say if you have an autoregressive image generator which generates small pictures (say 8x8) pixel after pixel, and each pixel is an int in [0,255]. Will your conclusion apply to this kind of sequential model? Will MLE work better than GAN?

Correct me if I am wrong or misunderstand your point.

**Limitations:**

I can completely understand this is a theoretical paper, but any experiment (even on toy/synthetic datasets) will further confirm the arguments made by this paper and examine to what extend all the assumptions stand.

**Strengths And Weaknesses:**

# Strengths

It’s well known that GANs don’t work so well in the domain of text compared to MLE, yet a thorough investigation is still lacking. The paper offers new and detailed insights into this phenomenon. Previous works explain this as the consequences of having discrete tokens, being not differentiable, sparse reward, optimization challenges, etc. This paper chose a different angle by showing how GAN and MLE are actually closely related.

The usage of “distinguishability” to analyze GAN is new.

The general reduction is useful and solid. It’s good to see asymptotic analysis.

The argument is backed with mathematics derivations. An essential algorithm is given. Assumptions are clearly given before claims.

The final conclusion is straightforward and clear.

Overall this paper is working on a topic with significance. People do care why GANs can/cannot work on text. This paper is making contributions and casting new insights into this problem.

# Weaknesses

The introduction of “distinguishability” is abrupt to me. See Q2.

The relation of distinguishability, likelihood(KL divergence) and JS divergence should be discussed further. See Q2 Q3.

It seems this paper focuses on next-token level GANs and not sentence level GAN. See Q4.

“GANs are overkill for NLP” seems too strong a claim, unless you can be very sure that you cover all types of GAN in NLP.

In general, I'm positive for this paper, but I still have some concerns that need to be addressed.

---

> ### Author Response · Authors · 2022-08-02
> **Response to review**
>
> We thank the reviewer for the very thorough feedback and suggestions. We provide answers to your questions below:
>
> 1. "*What’s the difference between 'distinguisher' and 'discriminator'?*". Thanks for the suggestion. Yes, what we call distinguisher for adversarial methods is commonly known as discriminator (e.g., as in original GAN) and critic (especially, in the context of more nuanced formulations based on some Integral Probability Metric, e.g.,  Sobolev GAN). We will unify these terms as you suggested.
> 2. "*Is this [that GAN training seeks to minimize distinguishability]  an observation, assumption or statement?*". Thanks for the opportunity to emphasize this important connection between distinguishability and GANs.  Distinguishability is indeed the objective that the adversarial approaches, such as GANs, seek to optimize. For example, as established in reference [1] below, GANs that are trained based on  Kantorovich metric, Fortet-Mourier metric, dual-bounded Lipschitz distance (or the Dudley metric), total variation distance, and kernel distance are all just specific formulations of the distinguishability criterion. Thus, in particular, our results hold for GNN formulations such as Wasserstein GANs [2], MMD GANs [3], Fisher GANs [4],  and Sobolev GANs [5], for an appropriately chosen family $F$ of distinguishers. For example, we obtain Wasserstein GANs, in its dual form, as a special case when $F$ is restricted to 1-Lipschitz functions in which case it can also be viewed as a special case of the so-called f-GANs [6, 7, 8]). Likewise, we obtain MMD-GANs when $F$ pertains to functions (kernels) defined over a ball in some Reproducing Kernel Hilbert Space [9]. We will make this clear based on your feedback.
> 3. "*Is 'distinguishability”' symmetrical (like JS) or asymmetrical (like KL)?*" Thanks for another great question that helps us elucidate the generality of our approach. Our distinguishers are asymmetric in the sense that in general 'q is distiguishable from p' is different from 'p is distinguishable from q'. This follows from our definition of distingishability of q from p as $d_q = \max_{f \in F} (E_q [f(x)] - E_p [f(x)])~.$  However, under our definition of distinguishability, we recover the notion of inverse probability metric (IPM) by letting $-f \in F$ for all $f \in F$. Clearly, in this case the notion of distinguishability becomes symmetric as it reduces to $\max_{f \in F} |E_q [f(x)] - E_p [f(x)]| ~.$ In fact, this explains in part, how the proposed framework lets us handle an extremely wide class of discrepancies, symmetric as well as asymmetric. Choosing an appropriate family $F$ of distinguishers immediately leads to the corresponding adversarial objective $\min_q d(q)$, where $d(q)$ is symmetric or asymmetric depending on $F$.
> 4. "*Does your argument work **only** on next-token level GAN?*" No. Due to space constraints, we post this as a separate comment below.
> 5. "*fundamental difference between GAN in text and vision?*" Yes, indeed, we can compute conditional likelihoods more efficiently for text data compared to images. The sequential nature of test data allows us to compute the normalization terms (and thus the partition function) on a token-by-token basis, thereby enabling us to distinguish the conditional next-token predictions instead of having to distinguish full sentences. Similarly, for low dimensional images (such as 8x8, or 4x4), conditional predictions are tractable and thus autoregressive modeling would allow for efficient training and sampling.  In contrast, MLE based autoregressive models such as PixelRNN [10] are typically slow for high dimensional real images. In such settings estimating the partition function is challenging, so alternative methods such as noise contrastive estimation, score matching, Langevin dynamics, and MCMC sampling in latent space that exploit connections between GANs and energy based models have been preferred [11, 12, 13, 14, 15].
>
> We thank the review, again, for their thoughtful feedback. We hope our response has sufficiently addressed all their questions and concerns, and if so, ask that they consider revising their score.

---

> > ### Author Response · Authors · 2022-08-02
> > **Detailed answer to question 4.**
> >
> >
> > Yes, our analysis extends to the sentence-level GANs as well. To make things concrete, we begin by noting that for any sentence $s = w_{1, s} w_{2, s} \ldots, w_{k_s, s}$, we can decompose the probability $q(s)$ (and likewise $p(s)$) in terms of conditional probabilities given the prefixes, namely,
> > $$q(s) = \prod_{i=1}^{k_s} q(w_{i, s}|w_{1,s}, \ldots, w_{i-1, s})~.$$
> >
> > Then, distinguishability can be written as
> > \begin{eqnarray*} &&\max_{f \in F} E_q [f(s)] - E_p [f(s)]\\
> > & = & \max_{f \in F} \sum_s (q(s) - p(s)) f(s) \\
> > & = & \max_{f \in F} \sum_s \underbrace{\left[\prod_{i=1}^{k_s} q(w_{i, s}|w_{1,s}, \ldots, w_{i-1, s})  - \prod_{i=1}^{k_s} p(w_{i, s}|w_{1,s}, \ldots, w_{i-1, s}) \right]}_{r(s)} f(s)
> > \end{eqnarray*}
> >
> > Thus, one can define a sentence level GAN that takes the entire sentence $s$ and implements a discriminator as well as a sequential component (such as LSTM/RNN) each for modeling (conditional) distributions $p$ and $q$. It can then produce a single score based on $s$ (more specifically, using $r(s) = q(s)-p(s)$ and $f(s)$).  Thus, unlike, next-token level GANs, $f(s)$ is computed only once (i.e., after processing the entire sentence). As you mentioned, this kind of GAN suffers from critical issues such as difficulties with sampling.
> >
> >
> > In contrast, we can still follow the general efficient reduction from section 6 in the paper for MLE that exploits step-wise weak distinguishers (recall that we do not need these distinguishers to be optimal). However, without step-wise distinguishers are not available, direct MLE estimation at the sentence level would itself be computationally demanding (due to combinatorial issues), and susceptible to high variance.
> >
> > In summary, the benefits of our approach stem from an efficient  reduction that leverages weak distinguishers for MLE based training. This, in spirit, is akin to boosting, where weak models (such as one-level decision trees, or stumps) can be sequentially combined to achieve a strong ensemble classifier, efficiently, compared to fitting a single optimal decision tree directly (which is known to be hard).  Our analyses establish that a wide class of adversarial objectives, including those prominent in NLP, can similarly be trained way more efficiently using MLE models instead.

---

> > > ### Author Response · Authors · 2022-08-02
> > > **Additional References cited in Response**
> > >
> > >
> > > [1] B. Sriperumbudur, K. Fukumizu,  A. Gretton, B. Scholkopf, and G. Lanckriet. On the empirical estimation of integral probablity measures (Electronic Journal of Statistics, 2012).
> > >
> > > [2] Martin Arjovsky, Soumith Chintala, and L. Bottou. Wasserstein Generative Adversarial Networks (ICML 2017).
> > >
> > > [3] Y. Mroueh and T. Sercu. Fisher GAN (NeurIPS 2017).
> > >
> > > [4] C.-L. Li, W.-C. Chang,Y. Cheng, Y. Yang, and B. Poczos. MMD GAN: Towards Deeper Understanding of Moment Matching Network (NeurIPS 2017).
> > >
> > > [5] Y. Mroueh, C.-L. Li, T. Sercu, A. Raj, and Yu Cheng. Sobolev GAN (ICLR 2018).
> > >
> > > [6] G. Biau, M. Sangnier, and U. Tanielian. Some Theoretical Insights into Wasserstein GANs (JMLR, 2021).
> > >
> > > [7] S. Nowozin, B. Cseke, and R. Tomioka. f-GAN: Training Generative Neural Samplers using Variational Divergence Minimization (NIPS 2016).
> > >
> > > [8] J. Song and S. Ermon. Bridging the Gap Between f-GANs and Wasserstein GANs (ICML 2020).
> > >
> > > [9] M. Binkowski, D. Sutherland, M. Arbel, and A. Gretton Demistyfying MMD GANs (ICLR 2018).
> > >
> > > [10] A. van den Oord, N. Kalchbrenner, and K. Kavukcuoglu. Pixel Recurrent Neural Networks (ICML 2016).
> > >
> > > [11] B. Dai, Z. Liu, H. Dai, N. He, A. Gretton, L. Song, and D. Schuurmans. Exponential family estimation via adversarial dynamics embedding (NeurIPS 2019).
> > >
> > > [12] S. Zhai, W. Talbott, C. Guestrin, and J. Susskind. Adversarial fisher vectors for unsupervised representation learning (NeurIPS 2019).
> > >
> > > [13] C. Finn, P. Christiano, P. Abbeel, and S. Levine. A connection between generative adversarial networks, inverse reinforcement learning, and energy-based models (arXiv:1611.03852, 2016).
> > >
> > > [14] T. Che, R. Zhang, J. Sohl-Dickstein, H. Larochelle, L. Paull, Y. Cao, and Y. Bengio. Your GAN is Secretly an Energy-based Model and
> > > You Should Use Discriminator Driven Latent Sampling (NeurIPS 2020).
> > >
> > > [15] Y. Song and D. Kingma. How to Train Your Energy-Based Models (arXiv: 2101.03288, 2021).

---

> > ### Comment · Reviewer_UUT1 · 2022-08-09
> > **Thanks for response.**
> >
> > Thank you for your detailed response. The explainations can very well enhance the paper content and address concerns. I hope the discussions can be reflected in your revised paper version, and also improve the writing and organization of the paper. Hopefully after that you can make your points unambiguously clear and eaiser for readers to understand. I revised my score and changed it from 5 to 6.

---

> > > ### Author Response · Authors · 2022-08-09
> > > **Thanks for your response.**
> > >
> > > Many thanks for your engagement in the discussion! We're grateful for your support, and will include your suggestions in the final version.

---

### Official Review · Reviewer_jtPN · 2022-07-11

**Rating:** 7
**Confidence:** 3
**Soundness:** 3 good
**Presentation:** 4 excellent
**Contribution:** 3 good

**Summary:**

This paper argues that training a GAN, which minimizes distinguishability between a learned and actual distribution, and maximizing likelihood, are often equivalent for NLP tasks. They do this by (i) giving a case where minimizing distinguishability is not the same as maximizing likelihood due to limitations in the set of possible generates, then (ii) showing how training a GAN is equivalent to maximizing likelihood for n-gram models, then (iii) showing how training distinguishing reduces log loss, and finally (iv) showing a polynomial-time reductions from a distinguishing distribution to an MLE for sequence models.

**Questions:**

* [140-141]: Is it the case that likelihood-based models generated text that could be easily distinguished from humans? I think fake-news detection, etc. has been an issue for at least 5 years?
* What is novel about the next-token distinguisher? Is this not a direct application of the standard GAN objective for classification to sequence modeling?

**Limitations:**

N/A.

**Strengths And Weaknesses:**

Strengths
* The paper gives good theoretical motivation for an empirical phenomenon that, to the best of my knowledge, wasn't well understood. In particular, previous explanations argued that
* The exposition in the paper is great; the authors help motivate with simple cases where distinguishability + MLE are different, then show how they're the same in an intuitive and realistic case, then provide general theory
* The explicit polynomial time reduction seems like a strong theoretical contribution.

Weaknesses
* The main weakness is the lack of empirical validation; in particular, it would've been nice to see if GANs actually do end up at the MLE in simple cases, and whether the rate at which they do is slower, as argued in the paper.

---

> ### Author Response · Authors · 2022-08-02
> **Response to review**
>
> We are grateful for the thoughtful feedback and the appreciation of the contributions of the paper. In particular, we are glad the reviewer appreciated the contribution of the polynomial time reduction. Answers below:
>
> - "*The main weakness is the lack of empirical validation; in particular, it would've been nice to see if GANs actually do end up at the MLE in simple cases, and whether the rate at which they do is slower, as argued in the paper."*. Please see the empirical validation we've added in the general comment section above. This is a first step towards the validation you propose, which requires addressing some additional issue, e.g., how to verify whether models with potentially different parametric representations (e.g., architectures) correspond to the same MLE solution. In other words, both could achieve a similar LL on a finite sample of data but not correspond to the same model.
>
> - "*[140-141]: Is it the case that likelihood-based models generated text that could be easily distinguished from humans? I think fake-news detection, etc. has been an issue for at least 5 years?*" Thank you for this interesting question. This was the case not too long ago, before the advent of Transformers and other very large NLL-trained language models. The discrepancy between quality and diversity of pre-Transformer NLG models and the ability of humans to pick up on each of these is discussed in detail in Hashimoto et al., 'Unifying Human and Statistical Evaluation for Natural Language Generation'.
>
> - "*What is novel about the next-token distinguisher? Is this not a direct application of the standard GAN objective for classification to sequence modeling?*" In short, yes, the principle behind the distinguishers we use here is very similar to (albeit more general than) the discriminator in usual GAN training. The novelty is not in using this principle to train a language model (in fact, we expressly advocate against doing so, see general comments above), but its use to formally prove the polynomial time reduction between training this adversary and fitting a MLE model.

---

### Official Review · Reviewer_vzAW · 2022-07-17

**Rating:** 3
**Confidence:** 3
**Soundness:** 3 good
**Presentation:** 3 good
**Contribution:** 2 fair

**Summary:**

The submission broadly aims at providing arguments for why GANs have not seen much success in sequential generation, as in language, but maximum likelihood models have.

The submission is entirely theoretical, demonstrating that for a reasonable notion of “distinguishability” of model samples from true samples, one can derive a method to minimize the empirical negative log-likelihood using the output of a classifier that can discriminate (with some effectiveness at least) the model samples from true samples. This derivation connects “distinguishability” (for example, predictions from an adversarial discriminator) with log-likelihood maximization. For sequential models, one can adapt the same procedure to conditional distributions over sequence items, yielding an algorithm reminiscent of boosting, where (possibly-) weak “distinguishers” can be used at each stage to further minimize negative log-likelihood. A run-time analysis of this algorithm is provided.

**Questions:**

1. Why is Algorithm 1 empirically non-validatable at this stage?

2. How does the recommendation in the paper to perform NLL optimization relate to observations in [4, 5]?

3. Can the method be modified to accommodate unbounded critics?

**Limitations:**

The authors have discussed potential negative societal impacts.

**Strengths And Weaknesses:**

I think some of the phrasing might be a bit misleading to some readers. The abstract suggests that the difference between maximizing likelihood for an explicit model and training an implicit model such as a GAN is “largely artificial”. Having read the paper, I do not find the equivalence this claim seemed to promise: it turns out that the main argument tying the two together is that one can use a differentiating-signal between two distributions to improve log-likelihood for an explicit model. L49-51 claims that GAN training is a “roundabout way of maximizing likelihood on observed data”, but in my reading, the submission does not really substantiate this — it is only shown that a current log-likelihood may be improved if one uses an effective adversary’s predictions to push down probability density at points distinguishable from the true samples by the adversary. In my view, this does not really imply that GAN training is maximum likelihood in disguise in any meaningful sense as seems suggestive in the submission, rather it only suggests that maximum likelihood training may be conducted by using (even weak) adversaries.

The lack of empirical support is somewhat disappointing, after deriving the algorithm. Algorithm 1 seems implementable, as long as one can design a $g$ that can handle multi-dimensional inputs (or $N$ of them), since Lemma 3 seems to suggest that the normalization is doable by only summing over $|\mathcal{V}|$ terms. Is there something I’m missing, in terms of practical application, which would explain the lack of empirical validation? Without experimental support, it is unclear if the connection in the paper is actionable: does it buy us anything? The conclusion suggests that the takeaway is that “in applications where it is natural to fit models by minimizing log-loss, it is indeed likely to be a more direct and efficient means of fitting a model”. So is the recommendation that one ought to use “distinguishers” to optimize NLL using the developed procedure? Or that we should directly attempt to optimize NLLs? What would be an example of an unnatural circumstance for fitting models with MLE (recall that for images, flow-based explicit models have been approaching GANs promisingly closely in terms of sample quality, for example [1]; the auto-regressive model PARTI from Google is highly convincing)?

It would be nice to have some clearer characterization of what is meant by “similar power” in F and Q.

In general, the question the submission seems to aim to tackle at a high-level (based on the title at the very least) might be too ambitious to settle. With all the moving parts in designing and training a model, such as architectural choices, optimization issues, issues of mode-collapse in generation, and the often unanticipated way they mesh  together (or don’t), the meta-aspects guiding efforts invested by the community on certain approaches over others depending on publicized initial results, pin-pointing reasons for why one class of methods might have seen less success than others is difficult unless a clear technical reason is identified. In my opinion, this submission does not really identify any such issue that clearly informs us why GAN training might be “overkill for NLP”. Once again, recall how the dominant narrative of “GANs are best for image generation, and nothing else comes close” has changed significantly in recent times to “flow, diffusion, and autoregressive models are starting to look really really good!”.

Typo in L349: “to” —> “two”


Some related work that are relevant:

 — L144-151 does not take into account variants of GANs, such as the WGAN-GP [2], that do not suffer from gradients vanishing on discrete-spaces. More generally, how does the discussion in the paper relate to cases like WGANs where the critic outputs are not in [0, 1]?

 — FlowGAN [3] is one relevant work. They report that, in their framework, optimizing for likelihood in a hybrid model results in poor sample quality but good likelihoods, and conversely, using adversarial training results in opposite trends. Related to this observation are the broader observations in [4, 5] about how improved likelihoods need not correspond to improved sample quality in practice for high-dimensional data.

Overall,

 — While the paper seems to have decent elements of originality in it,

 — and is very clearly written,

 — but does not seem to deliver sufficiently in order to be of much practical significance at the current stage.


[1] Glow, Kingma and Dhariwal, 2018

[2] WGAN-GP, Gulrajani et al., 2017

[3] FlowGAN, Grover et al., 2018

[4] Locally-connected transformations for deep GMMs, van den Oörd and Dambre, 2015

[5] A note on the evaluation of generative models, Theis et al., 2016.

---

> ### Author Response · Authors · 2022-08-02
> **Response to review (1/3)**
>
> - "*Title misleading / research question too ambitious to settle*". We take this point, and recognize that some of the phrasing of the paper might misrepresent its core message. In response to your comments, we have decided to rephrase/soften some of these statements, including the title, to make it clear that GANs are *often* (i.e., in the circumstances described above) overkill, but there still might exist specific situations in which they might work just fine.
> - "*Without experimental support, it is unclear if the connection in the paper is actionable: does it buy us anything?*". The goal of the paper is to provide further theoretical understanding on a well-known and empirically observed phenomenon: that MLE training for NLP models is far more successful than GAN training. Thus, its goal is *explanatory* rather *prescriptive*. That being said, we still think it is actionable, in the sense that it might prevent further futile efforts to `make GANs work for NLP'.  As discussed in the general comment above, our reductions are intended to serve a purpose similar to the polytime reductions routinely conjured in complexity theory: e.g., one can establish that a problem B (e.g., independent set) is NP-complete if it admits a polytime reduction to some already known NP-complete problem A (e.g., vertex cover). These reductions are not intended to necessarily design an (approximate) algorithm for problem B that invokes procedure for A, but rather to redirect the efforts on B toward a more fruitful setting by underscoring the hardness of computing B given what we know about its relation to A.
> - "*So is the recommendation that one ought to use “distinguishers” to optimize NLL using the developed procedure?*" As stated in the general comment above, Algorithms 1/2 are not intended as feasible optimization methods, but rather as algorithmic reductions. We emphatically do not recommend their usage in practice. In fact, the take-home message is the opposite: when NLL is directly efficiently optimizable (as is the case for NLP and other sequential data), using distinguisher/GAN-based methods instead is futile.
> - "*It would be nice to have some clearer characterization of what is meant by “similar power” in F and Q*". Thank you for pointing out this ambiguity. Here ``power'' refers to representational capacity. We will clarify this in the paper.
> - Regarding the lessons from FlowGAN, etc: Thank you for the oppportunity to elaborate on this. Regarding the lessons from FlowGAN and other references you cited, we completely agree that likelihood and 'quality' or 'realistic-ness' of sample do not always go hand in hand. In fact, one prominent example of this phenomenon  is the standard VAE objective [1] that tries to maximize a lower bound on the log-likelihood but results in poor sample quality compared to GANs. Turns out that, in such settings, distributional shift due to minimizing (regularized) distortion can be at odds with the perceptual quality of the samples [1, 2]. On the other hand, in effect, GANs  also take into account the KL divergence in the other direction, leading to comparatively much better quality of samples. Indeed, understanding the theoretical underpinnings of generative models with respect to their sample quality is an intriguing question that requires further analyses. In the context of present work, **our message here is not to claim at all that maximizing likelihood is universally better than adversarial methods or vice-versa, but to emphasize that for many problems, in domains like NLP, the two objectives often turn out to be equivalent *mathematically* via the notion of distinguishability and maximizing MLE could provide a more efficient (and stable way) of optimizing the common objective**. Surely, as you rightly mentioned, a more comprehensive investigation encompassing the interplay of several intriguing factors such as optimizing methods, network architecture, step sizes, momentum based acceleration, size of the models, etc. is very much needed to further our understanding of the relative pros and cons of different generative models in such scenarios. Our work should be seen as a stepping stone toward that endeavor. Based on your feedback, we will add a discussion on this along with the references you mentioned.

---

> > ### Author Response · Authors · 2022-08-02
> > **Response to review (2/3)**
> >
> > - "*this submission does not really identify any such issue that clearly informs us why GAN training might be 'overkill for NLP'*".  We acknowledge your point, and would like to point out in our defence that several complex factors are at interplay behind this phenomenon. The sequential nature of text data affords efficient estimation of important quantities (e.g., the partition function), and this bestows MLE based methods, in effect, with the ability to distiguish the conditional next-token predictions rather than having to distinguish full sentences for the class of problems we discussed here. At the same time, MLE based autoregressive models allow for efficient and stable training (and sampling) in this setting. This is in contrast, e.g., to typical vision settings where the MLE based autoregressive models end up being considerably less efficient due to high dimensionality of the input images. We reiterate that there isn't a single issue that accounts for GANs being \textit{overkill} (i.e., unnecessarily complicated --- see our explanation on the choice of this word above), but rather a combination of factors, which we discuss in Section 1: GANs are conceptually and implementation-wise much more involved than MLE training, and much more prone to unstable training, a price that might be worth paying when the alternatives (like MLE) are unsuccessful -  e.g., MLE typically performs poorly in settings where model misspecification is a major issue -  which is certainly not the case for the NLP problems  we focused on in this work. It is our hope that this paper fosters further exciting work toward unraveling the effect of such issues.
> > - "*Once again, recall how the dominant narrative of “GANs are best for image generation, and nothing else comes close” has changed significantly in recent times to “flow, diffusion, and autoregressive models are starting to look really really good!”*". Thanks for underscoring this point.  Certainly, the narrative evolves with evidence; and particularly in the context of NLP applications, there is an overwhelming empirical evidence that GANs have not been nearly as successful as they have been in several other applications. In that sense, our work is an attempt to establish a rigorous mathematical connection between two prominent classes of generative models via the notion of distinguishability. Diffusion models have only recently begun to gain attention within the NLP community (e.g., Austin et al, NeurIPS 2021) and certainly present an exciting avenue.  Interestingly, the analysis presented here could help unravel the connections between diffusion models and maximum likelihood based models for specific class of problems (such as the log-linear models) via their similarities/interpretations as energy-based models (Song and Kingma, arXiv: 2101.03288).
> > - "* L144-151 does not take into account variants of GANs [...]*". Thank you for this observation.
> >  The discussion in L144-151 concerns the need for a differentiable discriminator, which is still the case for WGAN (and all its variants). It is true that Wasserstein distance-based objectives address that issue of vanilla GANs, but the issue of differentiability remains (and indeed, has been approached with various post-hoc methods like Gumbel-softmax, etc). We would also like to point out that in practice, WGAN (but not WGAN-GP) resorts to, and recommends, weight clipping to make them lie within a compact space. Then, our analysis carries over via the family $F'$ of distinguishers to include appropriately scaled critics $f'$ as well as $-f'$.   Our framework is also flexible enough to accommodate variants such as WGAN-GP. Recall that the WGAN-GP formulation can be expressed in our notation as: $ E_q [f(x)] - E_p [f(x)] + \lambda  E_r [(\|\nabla_x f(x)\|_2 - 1)^2] $ . This can be viewed as Lagrangian relaxation of the following hard objective for $\epsilon > 0$ as: $E_q [f(x)] - E_p [f(x)]$ subject to $ E_r [(\|\nabla_x f(x)\|_2 - 1)^2] < \epsilon $. Distinguishability can then be readily be expressed as:
> >
> > $\max_{f : E_r [(\| \nabla_x f(x) \|_2 - 1)^2]  <  \epsilon} E_q [f(x)] - E_p [f(x)] $,
> >
> > which is clearly of the form this work deals with, i.e., $\max_{f \in F} E_q [f(x)] - E_p [f(x)]$.

---

> > > ### Author Response · Authors · 2022-08-02
> > > **Response to review (3/3)**
> > >
> > >
> > > - "* How does the recommendation in the paper to perform NLL optimization relate to observations in (O\"{o}rd and Dambre, 2015) and (Theis et al., 2016.)?*" The recommendation in this paper agrees with (Theis et al)  'Our results demonstrate that for generative models there is no one-fits-all loss function but a proper assessment of model performance is only possible in the
> > > the context of an application',  and (van den O\"{o}rd and Dambre) 'In this work we introduced new ways of modeling images
> > > with deep GMMs by using locally-connected transformations. These transformations efficiently exploit the fact that correlations in images are stronger between pixels that are closer to each other. This allows much faster training and
> > > less overfitting'. Namely, this paper establishes mathematical equivalence between maximizing likelihood and minimizing distinguishability for a wide class of NLP settings, and thus recommends NLL \textbf{only} for applications when it is mathematically equivalent but more efficient and direct to compute compared to a broad class of adversarial methods. In particular, we do not make any claims about settings where the two approaches clearly optimize different objectives. Thank you for bringing our attention to these works. We will position the implications of the proposed work appropriately with respect to these important references in the revised version.
> > > - "*Can the method be modified to accommodate unbounded critics?*". The conclusions from this work apply to any training method based on distinguishability, which includes IPMs, Wasserstein distances, TVD, etc, and is not limited to methods without outputs in $[0,1]$. Moreover, variants such as WGAN-GP can also be investigated within the proposed framework as described earlier.  We will include these references, and add a discussion on this.
> > >
> > >
> > > Thank you for your constructive feedback. Hope our response has sufficiently addressed your concerns and the same would be reflected in your revised scores.

---

> > > > ### Author Response · Authors · 2022-08-09
> > > > **Revise score**
> > > >
> > > > Would you consider revising your score in light of our substantial modifications, including experiments we have added (see the general comments above) ?

---

> > > > ### Author Response · Authors · 2022-08-09
> > > > **Request your engagement on our response**
> > > >
> > > > Thanks, again, for your constructive feedback! Acting on your comments and questions has helped us clarify and reinforce some strengths of this work along multiple dimensions, including,
> > > >
> > > > (1)  Technical: see, e.g., the mathematical justification on generality of the proposed framework, including how these analyses apply to methods such as WGAN-GP; and
> > > >
> > > > (2) Empirical: see, e.g., the general comment above where we show 'boosting' underpinning Lemma 6 does indeed improve likelihood (reduces NLL) as stated, and that the empirical difference is indeed lower-bounded by the gap predicted by theory.
> > > >
> > > > Therefore, we would be grateful if the same could be acknowledged, and translated into a revised score.

---

> ### Comment · Reviewer_vzAW · 2022-08-09
> **Post-rebuttal update**
>
> Thanks for clarifying some of the points!
>
> _“…we have decided to rephrase/soften some of these statements, including the title, to make it clear that GANs are often (i.e., in the circumstances described above) overkill, but there still might exist specific situations in which they might work just fine.”_ — Have these changes been made in the main document? How is the title going to change?
>
> _“…goal is explanatory rather prescriptive. That being said, we still think it is actionable, in the sense that it might prevent further futile efforts to `make GANs work for NLP’. / when NLL is directly efficiently optimizable (as is the case for NLP and other sequential data), using distinguisher/GAN-based methods instead is futile.”_ — This is actually one of the reasons I’m hesitant to promote this work. It’s not clear to me at all from the specific discussion in the paper that anyone ought to feel dissuaded from trying to make GANs work for NLP just yet.  It’s not obvious to me that reduction arguments of the type used in the submission ought to have significant bearing in deep learning practice, given the many interacting components behind getting something to work. Different formulations of the same asymptotic objective might cause one to take very different roads, and when the landscape is treacherous, one can end up in very different places. At the end of the day, empiricism ought to determine best practice, unless a very precise technical reason is identified behind the _infeasibility_ of a class of methods.
>
> My view after reading the submission was that it needed a do-over in several places, as well as a change in title, to avoid the potential risk of sparking an inclement GAN-winter in NLP, where perhaps we are only waiting for some major technical breakthrough. It appears from the rebuttal that some changes have been made, but it’s not clear to me where and how. I’m still of the view that this submission ought to be reviewed from scratch before being disseminated.

---

> > ### Author Response · Authors · 2022-08-09
> > **Thank you - please see the revised version.**
> >
> > Thank you for your response!  Our purpose is to attract greater attention of the community to an important area of research, and we believe the conceptual and mathematical contributions of this paper would be broadly useful in the context of generative models.  We've edited the paper at several places including the title (changes indicated in orange) - please see the revised version - to make sure that your reservations are accounted for.  We hope these changes address your major concerns, and the same is reflected in your stronger support during the reviewer-area chair discussions.

---

### Official Review · Reviewer_broM · 2022-07-30

**Rating:** 4
**Confidence:** 2
**Soundness:** 2 fair
**Presentation:** 2 fair
**Contribution:** 2 fair

**Summary:**

This paper analyzes why GANs underperform MLE in natural language generation tasks. The authors argue that minimizing KL-divergence like MLE is a more efficient approach compared with minimizing the same distinguishability criteria in adversarial models. The authors also propose that minimizing distinguishability can be regarded as boosting likelihood for certain families of models including n-gram models and neural networks with a softmax output layer.

**Questions:**

I have included my questions in the weaknesses part.

**Limitations:**

The authors have adequately addressed the limitations and potential negative societal impact of their work.

**Strengths And Weaknesses:**

Strengths:

1. It’s essential and meaningful to give a theoretical analysis on why text GANs fall short. The authors try to connect the minimization of distinguishability and the boosting of the MLE training objective, which is an interesting perspective.

Weaknesses:

1. The organization of this paper should be largely improved. It’s hard for me to follow the detailed derivation because there is always a gap between consecutive formulas. Also, as a paper focusing on text GANs, most parts are about general GANs which are not directly related to texts. For example, the authors mention GANs based on the n-gram model in Section 3. But the sequential formulation of q(x) where $x=w_1 w_2 \cdots w_t$ is not applied to the following derivation in Section 4 and 5. I feel that many parts of this paper are used to analyze general GANs (such image GANs), which are divergent from the title.
2. As mentioned in Section 2, the non-differentiable problem of text GANs is always solved by policy gradient or Gumbel-Softmax approximation. I wonder whether the following theoretical analysis consider this step because it plays an important role in the performance of text GANs from the existing works.
3. Since this paper proposes a specific algorithm (i.e., Algorithm 1 in Section 6), the authors should conduct an experiment (at least on synthetic data) to show the effectiveness of the proposed algorithm.

---

> ### Author Response · Authors · 2022-08-02
> **Response to review.**
>
> We thank the review for the feedback. Answers below:
>
> - "*It’s hard for me to follow the detailed derivation because there is always a gap between consecutive formulas*". Thank you for the suggestions for improving the presentation. We will edit the layout to prevent this from happening.
> - "*paper focusing on text GANs [...] but the sequential formulation of q(x) where  is not applied to the following derivation in Section 4 and 5*". Indeed, the results of 4 and 5 apply to general log-linear models, which are common (but not exclusive) to NLP and text data. After deriving these general results in Section 5, we tailor them to the sequential setting in Section 6.
> - "*I feel that many parts of this paper are used to analyze general GANs [...] which are divergent from the title*". As stated above, many of our results are more general, but their *implications* for NLP is the key takeaway message of this work. More concretely, showing such an equivalence between GAN and MLE training for this general class of models has substantially  different implications for text data (where MLE is typically easy and efficient) than for image data (where MLE has traditionally, and until very recently, been much more challenging). The title and discussion of this paper focus on the former.
> - "*the non-differentiable problem of text GANs is always solved by policy gradient or Gumbel-Softmax approximation. I wonder whether the following theoretical analysis consider this step*". We agree that these tricks are crucial component of making GANs for text data work (and indeed we discuss them in Page 4). However, what matters for our analyses is that the models are trained to minimize a certain objective (e.g., distinguishability), but the \textit{mechanics} of how that is achieved is less important and does not play a role in the results.
> - "*Since this paper proposes a specific algorithm [...] should conduct an experiment to show the effectiveness of the proposed algorithm*".
> Please see the general comment on this above.

---

> > ### Author Response · Authors · 2022-08-09
> > **Experiments**
> >
> > Would you consider revising your score in light of our modifications, especially the experiments we have done as you have suggested (see the general comments above) ?

---

### Author Response · Authors · 2022-08-02
**General Comments**

We thank the reviewers for their feedback and comments. We would like to address some common points raised by multiple reviewers.

- **On the applicability of Algorithm 1**. Multiple reviwers asked why there was no empirical validation of Algorithm 1 or the Lemmas. First, we would like to emphasize that the Algorithm we propose is not intended to be a practical one. It is an (polynomial-time) algorithmic reduction proving two problems are equivalent by showing how the solution of one can be used to solve the other. This is a common proof technique in the theory of computer science literature, and the resulting algorithms are rarely implemented -- if they are implementable at all. In this paper, the algorithmic reduction is a theoretical tool too, and we neither advise its use in practice nor we claim it as a practical contribution. In fact, its use would go against the main message of this work: that when MLE training is practical and efficient, GAN training is superfluous. That being said, based on the requests of the reviewers we have implemented a simple initial empirical validation of the reduction underpinning the algorithm (Lemma 3), which we discuss in the next point.

- **On the choice of title: overkill $\neq$ useless**. We would like to emphasize that our choice of work 'overkill' for the title does not imply a message that GANs never work for NLP or should never be used, but rather that they might be unnecessarily complicated (at least, compared to MLE, which is arguably much more straightforward and well-understood).


- **An empirical validation of the reductions**. We have implemented and experimented with an empirical validation of Lemma 3 in the following realistic (albeit simplified) setting. We train a pair of text generator and discriminator using a [publicly available implementation ](https://github.com/suragnair/seqGAN) of SeqGAN (Yu et al. AAAI 2017). The generator is pretrained by negative log-likelihood (NLL) minimization. During the adversarial phase of training, the generator is trained using policy gradient. After training, we compute the discriminators' generalized training advantage (Eq. (4), using finite-sample empirical approximations), and then create a new generator whose next-word logit predictions are modified according to Lemma 3. We compare the NLL of the original and 'boosted' generators across training epochs, and compare the difference between these to the theoretical lower bound of Lemma 3. The results below correspond to the default repo settings in terms of network capacities and language configuration (`max.~seq.~len.~=20`, `vocab=5000`). They show that the 'boosting' underpinning Lemma 6 does indeed improve likelihood (reduces NLL) as stated, and that the empirical difference is indeed lower-bounded by the gap predicted by theory. With the benefit of additional time, we intend to run these experiments on various training configurations to get a better understanding of the empirical behavior of this reduction for varying model capacities and initial performance (esp.~weaker generators with lower initial NLL).

|   Discrim. Epoch |  Gen. Epoch |   Gen. NLL |  'Boosted' Gen. NLL |  Improvement | Improvement Predicted by Bound |
|------------:|------------:|----------:|-----------:|-----------:|--------------:|
|           0 |           0 |   91.2756 |    91.1459 |   0.129694 |    0.00842083 |
|           1 |           1 |   91.2995 |    91.182  |   0.117515 |    0.00733052 |
|           2 |           2 |   91.3425 |    91.2238 |   0.118727 |    0.00834904 |
|           3 |           3 |   91.4    |    91.2872 |   0.112812 |    0.00683975 |
|           4 |           4 |   91.4663 |    91.3402 |   0.126077 |    0.00961482 |
|           5 |           5 |   91.5386 |    91.4103 |   0.12824  |    0.00769221 |
|           6 |           6 |   91.6125 |    91.4928 |   0.11971  |    0.00744601 |
|           7 |           7 |   91.6866 |    91.5565 |   0.130122 |    0.00836084 |
|           8 |           8 |   91.7617 |    91.6169 |   0.144865 |    0.00993902 |
|           9 |           9 |   91.8349 |    91.7039 |   0.131023 |    0.00818018 |
|          10 |          10 |   91.9048 |    91.7848 |   0.11997  |    0.00730441 |
|          11 |          11 |   91.9714 |    91.8493 |   0.122075 |    0.00844588 |
|          12 |          12 |   92.0345 |    91.8981 |   0.136373 |    0.00921325 |
|          13 |          13 |   92.0937 |    91.9648 |   0.128912 |    0.00832491 |
|          14 |          14 |   92.1491 |    92.0212 |   0.127937 |    0.00842633 |
|          15 |          15 |   92.2005 |    92.0755 |   0.125091 |    0.0079117  |
|          16 |          16 |   92.2481 |    92.1153 |   0.132838 |    0.0102334  |
|          17 |          17 |   92.2919 |    92.1581 |   0.133824 |    0.00773296 |
|          18 |          18 |   92.3321 |    92.2087 |   0.123441 |    0.00720734 |
|          19 |          19 |   92.3676 |    92.237  |   0.130602 |    0.00950763 |

---

### Meta-Review · Area_Chair_7yZA · 2022-08-30

**Recommendation:** Accept
**Confidence:** Less certain

**Metareview:**

In the context of text generation, the paper gives a theoretical argument that GAN objectives are equivalent to maximum-likelihood training when the generator and discriminator families are 'paired'. Reviewers generally felt that the perspective was interesting (broM, jtPN, UUT1) and the theory was insightful (jtPN, UUT1). Reviewer vzAW raises the concern that the original draft of this paper overclaimed throughout, but I feel this has been addressed well enough in a revision. Reviewers broM and vzAW felt empirical validation was lacking, but since the paper's focus is clearly theoretical I don't see this as preventing acceptance. Overall this paper is borderline but I feel that it's interesting enough to merit acceptance despite flaws.


**Award:**

No

---

### Decision · Program_Chairs · 2022-09-14

Accept